# Long-Term Geomorphological Evolution of the Mouth Bar in the Modaomen Estuary of the Pearl River over the Last 55 Years (1964–2019)

Zhiyuan Han [1,2,*], Huaiyuan Li [1,2], Hualiang Xie [1,2], Bing Yan [1,2] and Mingxiao Xie [1,2]

[1] National Engineering Research Center of Port Hydraulic Construction Technology, Tianjin Research Institute for Water Transport Engineering, M.O.T., Tianjin 300456, China; lhy07031@126.com (H.L.); xh-liang@foxmail.com (H.X.); yanbing@tiwte.ac.cn (B.Y.); crabsaver@163.com (M.X.)

[2] Key Laboratory of Engineering Sediment of Ministry of Transport, Tianjin Research Institute for Water Transport Engineering, M.O.T., Tianjin 300456, China

* Correspondence: tkshzy@foxmail.com; Tel.: +86-22-5981-2345-6416

**Abstract:** Based on mass bathymetric data and remote sensing data in the Modaomen Estuary, this study explored the long-term evolutionary characteristics of the mouth bar in the Modaomen Estuary of the Pearl River from 1964 to 2019. In the past 55 years, due to the impact of human activities, such as shoal reclamation and estuarine regulation in the Modaomen Estuary, the river mouth moved out of the shallow sea covered by several islands and faced the South China Sea directly. Therefore, the mouth bay became a siltation center in the estuarine region and expanded outwards, gradually evolving a geomorphic pattern with three shallow shoals and two distributary branches; a west branch as the main branch accompanied by a small east branch. Over the past decade, high-intensity sand dredging activities in the mouth bar have led to a considerable deepening of the water depth and a significant refinement of bed sediments, forming a discharge pattern of a wide and shallow channel flowing into the sea. Therefore, the evolutionary characteristics of the mouth bar have become abnormal in recent years, so additional field bathymetric data and hydrological data are required for further research regarding the subsequent evolution of the mouth bar, against the background of a significant reduction of sediment discharge and high-intensity human activities.

**Keywords:** mouth bar; geomorphological evolution; human activities; Modaomen Estuary; Pearl River



## 1. Introduction

Estuarine regions with strong land–ocean interactions are complicated; usually affected by fluvial dynamics, tidal dynamics, wave dynamics, and morphological evolution; the ecological environment of such regions is also sensitive and fragile [1–3]. In recent decades, with the development of the estuarine regions, human activities, such as upstream dam construction, large-scale land reclamation in estuarine shoals, sand dredging, as well as the construction of ports, navigation channels, and sea-crossing bridges, have notably altered the river discharge and sediment input, underwater topography, wave and tidal dynamic conditions. Consequently, the ecological environment has become more fragile and the sustainable management of the estuary has become increasingly challenging [1,4–9].

The mouth bar, as an accumulative underwater geomorphic unit in the estuarine region, is primarily formed due to the deposition of river sediments at the river mouth [3,10–12]. Mouth bars are well developed in estuarine regions with large rivers and sediment discharge, such as the Nile, Ebro, Yangtze River, Yellow River, Pearl River, and Liaohe Estuaries among others [5,13–17]. The morphological evolution of the mouth bar, which is influenced by runoff, tides, waves, sea-level rise, storm surge, and human activities, is an important research topic regarding land–ocean interactions in the coastal zone (LOICZ) [4,5,18]. The research topic involves estuarine regulation, navigation development, water resources management, and other fields [19–22].

The Pearl River, which is the largest in South China, flows into the South China Sea. The Pearl River Estuary (PRE) is composed of the Pearl River Delta (PRD) and estuarine bays [23]. The Modaomen Estuary is located at the southernmost end of the PRD, and is the main flow path of the Pearl River into the sea, especially during flood events. With the largest volume of river and sediment discharge coming from the Pearl River, a typical mouth bar has developed in the river mouth of the Modaomen Estuary. Since the 1960s, large-scale estuary regulation, sand excavation, navigating channel development, and other such human activities have been undertaken in the Modaomen Estuary [24–26]. Significant changes have taken place in the estuary and notably have influenced the morphological form of the mouth bar. The morphological evolution of the mouth bar in the Modaomen Estuary could influence the water supply safety of local cities, such as Macao, Zhuhai, and Zhongshan [25]. The existence of the mouth bar also hinders the navigation of big ships to the open sea, and some previous studies have mainly focused on the impact of human activities such as estuarine regulation on the estuarine tidal dynamics and salt water intrusion [26,27]. Studies on the mouth bar have also focused on the short-term morphological evolution after the regulation project [27,28]. However, this study on the morphological evolution of the mouth bar in the Modaomen Estuary spanning half a century is unique. Based on the measured topographic data of Modaomen Estuary for more than 50 years, this study attempts to address a crucial research gap by focusing on the long-term evolution process and mechanisms of the mouth bar. This research work can deepen our understanding of the evolution of the mouth bar in estuarine regions and be helpful for the sustainable management and protection of the PRE.

## 2. Study Area

The Pearl River is the second largest river in China and forms a large-scale river delta—PRD—and two large estuarine bays—Lingding Bay and Huangmao Bay—in the central Guangdong Province; the river has eight outlets into the sea. The Modaomen Estuary, as the only outlet directly into the South China Sea, is the main flow path of the Pearl River into the sea with a large river volume and sediment discharge. Makou hydrological station is the main control station upstream of the Modaomen Estuary, with an annual runoff of 224.4 billion $m^3$ and an annual sediment load of approximately 61.29 million tons, accounting for 76% and 85% of the total amount of the Pearl River, respectively. The total runoff and sediment load at Makou station in flood season (April–September) accounts for 76.5% and 94.5% of the whole year, respectively. The annual runoff and sediment load of the Modaomen Estuary is the largest of the eight outlets, accounting for 35–40% of the total amount of runoff and sediment load from Makou station.

The upper boundary of the Modaomen Estuary, which is located in Denglong Hill, is connected to the Modaomen Channel of the Xijiang River (see Figure 1b). Below Guading Point, the Modaomen Channel is divided into two runoff channels—the Jiaobeisha and Hongwan Channels—controlled by a fixed boundary. The Jiaobeisha Channel is the main channel of the Modaomen Estuary into the sea. It enters the sea near Shilanzhou, and a mouth bar has developed outside the river mouth. The Hongwan Channel is a small branch which flows into the sea near Macao. Longshiku is a tide-dominated channel which is located to the west of the mouth bar.

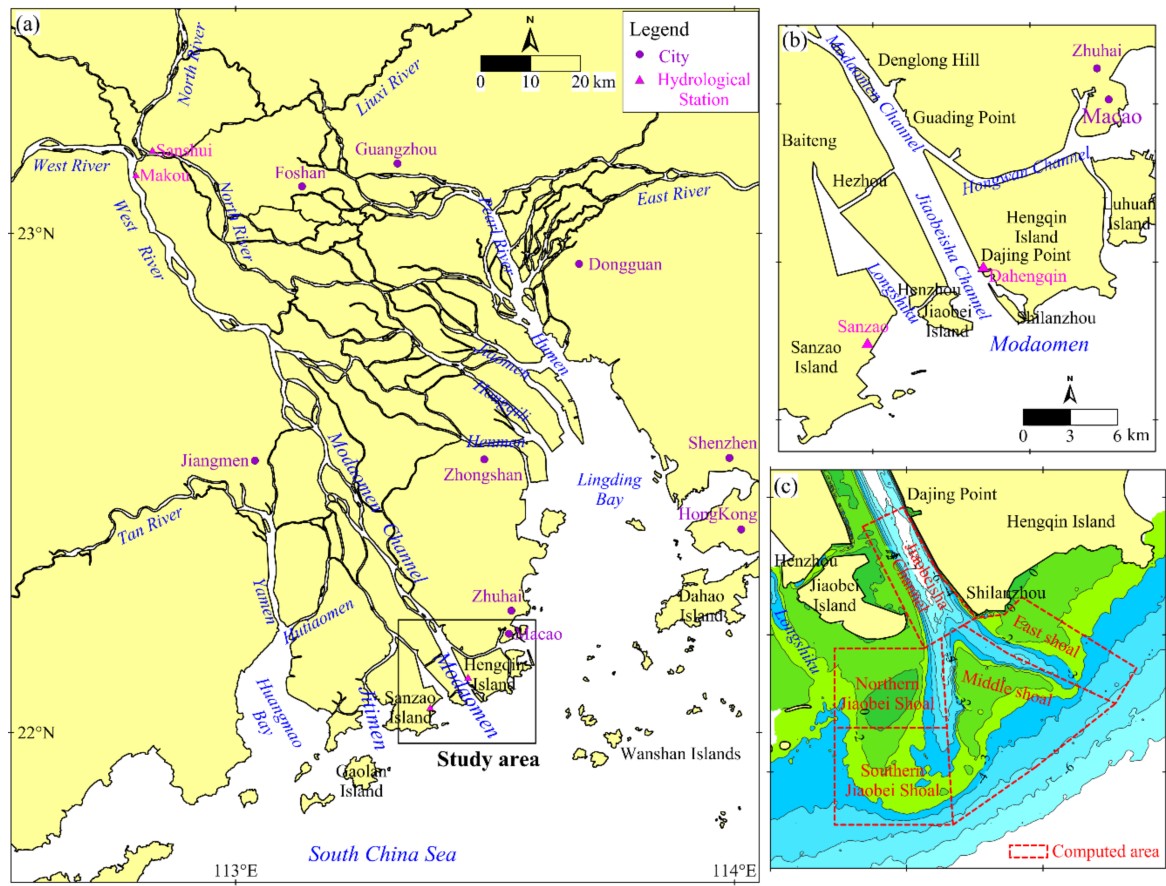

**Figure 1.** Sketch map of the PRE (**a**); the Modaomen Estuary (**b**) and computed area of the mouth bar (**c**).

The Modaomen Estuary is a fluvial-dominated estuary, where the tide is relatively weak with a tidal range of approximately 1.1 m at the Sanzhao station and a tidal type of irregular semidiurnal tides. The ratio of the multiyear average river discharge to the flood tide discharge is approximately 5.8 [28]. The tide type in the Modaomen Estuary belongs to the irregular semidiurnal mixed tide pattern. The tidal current flows reciprocally with the dominant direction being NW–SE in the Jiaobeisha Channel and flows rotationally outside the river mouth. The wave types outside the river mouth are mainly swells, with the dominant directions of SSE and SE. The biggest waves mostly occur during a period affected by typhoons.

## 3. Materials and Methods

### 3.1. Materials

Two China charts (chart No. 10722 and 10741, measured in 1964) which cover the Modaomen Estuary with a map scale of 1:50,000 were collected for this study. Data from nine bathymetric surveys, measured in March 1983, April 1994, May 2000, September 2005, March 2008, May 2010, July 2016, July 2017, and December 2019 with a map scale of 1:5000, and the 1985 National Height Datum, were provided by the Waterway Bureau of Guangdong Province. The plane coordinate system of the nine bathymetric data sets was the Beijing 1954 coordinate system.

Six remote sensing (RS) images, including one Landsat MSS image, four Landsat TM images and one Landsat OLI image, with separate spatial resolutions of 78, 30, 30, 30, and 30 m, were acquired for 2/11/1978, 24/11/1988, 30/11/1995, 1/11/2000, 4/3/2008, and 30/11/2019, respectively. All the RS images were acquired from the Geospatial Data Cloud website (http://www.gscloud.cn/s, accessed on 26 November 2021). The annual

runoff and sediment load data collected since 1959 for the Xijiang River (Makou station), the main tributary of the Pearl River, were provided by the Guangdong Hydrological Bureau. The seabed sediment data in the Modaomen Estuary measured in 2006 and 2019 were acquired from Sun Yat-Sen University and the Guangdong Waterway Bureau, respectively.

*3.2. Methods*

3.2.1. Bathymetric Data Processing

The two China charts were digitalized as bathymetric data using ArcGIS software. The coordinate system of the two charts was converted to the China Beijing 1954 Coordinate System with a central longitude of 114° E and the datum of water depth was corrected to the 1985 National Height Datum. Then, digital elevation models (DEM) of the Modaomen Estuary were generated with 10 m × 10 m grids using the Kriging interpolation method. Contour maps of different isobaths and planar maps of scouring and silting from 1964 to 2019 were drawn using ArcGIS software.

Finally, the mouth bar in the Modaomen Estuary was divided into five computed areas including the Jiaobeisha Channel (JC), northern Jiaobei Shoal (NJS), southern Jiaobei Shoal (SJS), middle shoal (MS), and east shoal (ES) (for locations see Figure 1). The depth and volume of the Jiaobeisha Channel (JC) below the 0 m isobaths and the other four computed areas above the −6 m isobaths were calculated using the cut-fill process of ArcGIS software.

3.2.2. Remote Sensing Image Processing

The shorelines for 1978, 1988, 1995, 2000, 2008, and 2019 were extracted from the six remote sensing images. According to the images, the extraction methods were as follows: (1) the preprocessing of the images, including the radiometric correction and geometric registration, was performed by the related tools in ENVI software, following which the DN (digital number) values of the images were changed to radiance values. The coordinate systems of the RS images were converted to the Beijing 1954 Coordinate System, and more than 10 ground control points with a root mean square error of less than 0.5 pixels for each image were selected to ensure the geometric accuracy of all images; (2) the artificial shorelines were directly extracted based on visual interpretation by the ENVI software, such as embankments of reclamation and port shorelines, among others; (3) the natural shorelines were extracted based on the normalized difference water index (NDWI) method [29]; (4) finally, all shorelines were placed on the same map using the ArcGIS software and the reclamation area was calculated for different decades.

## 4. Results

Based on the bathymetric data of the Modaomen Estuary from 1964 to 2019, spatiotemporal changes of the mouth bar and southern Jiaobeisha Channel were analyzed. Subsequently, the long-term evolutionary characteristics of the mouth bar were revealed.

*4.1. Planar Changes of the Mouth Bar*

According to the bathymetric data, the topography in different years (see Figure 2), and the isobaths changes of −2 m, −3 m, −4 m, and −5 m (see Figure 3) were drawn to study the planar changes of the mouth bar of the Modaomen Estuary from 1964 to 2019.

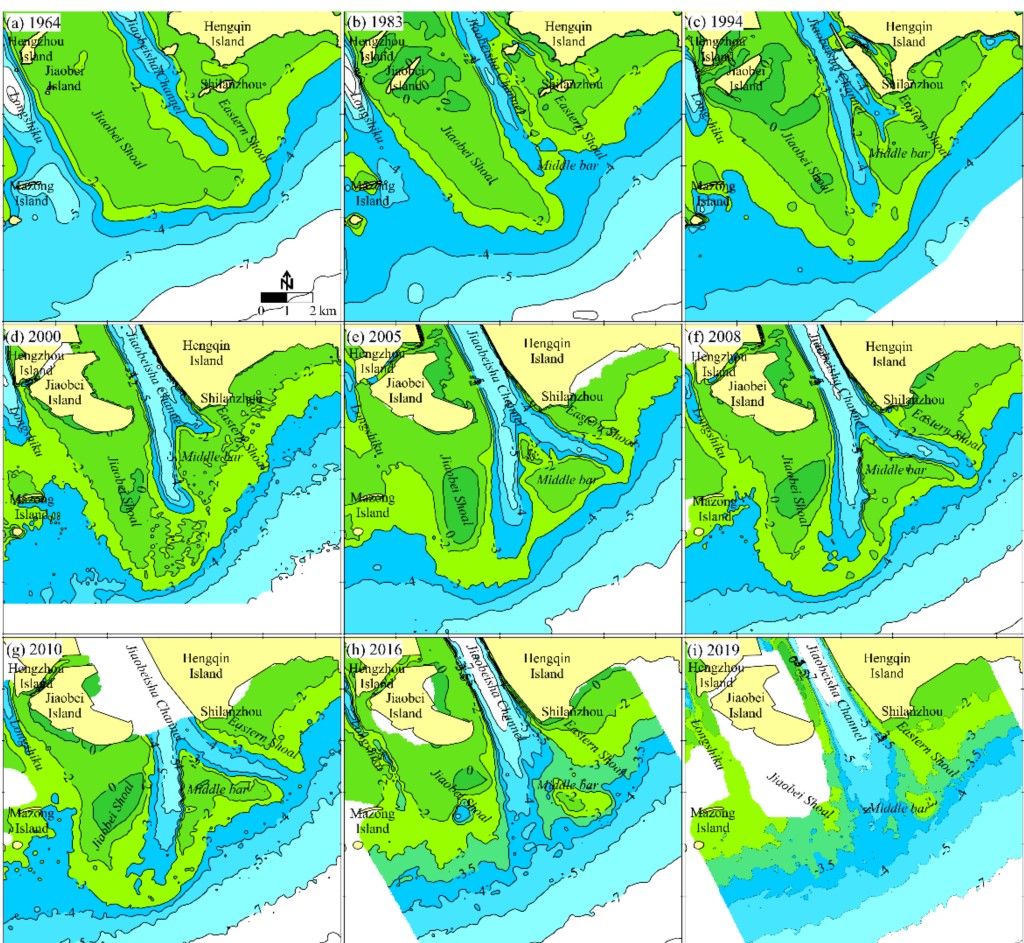

**Figure 2.** Underwater topography of the mouth bar from 1964 to 2019. (**a–i**) are the topography measured in 1964, 1983, 1994, 2000, 2005, 2008, 2010, 2016, and 2019, respectively.

4.1.1. Planar Changes from 1964 to 1983

According to the underwater topography in 1964 and 1983 (Figure 2a,b), the landform of 'two shoals with one branch' appeared outside the river mouth of the Modaomen Estuary. The Jiaobei Shoal and the eastern shoal were clear, with a water depth of less than −2 m. The south end of the Jiaobei Shoal extended eastward. The extension direction of the branch at the north of the mouth bar was SE, and there was no obvious distributary shoal outside the deep trough. The water depth in the Jiaobei Channel to the north of Shilanzhou was less than −5 m.

From 1964 to 1983, in the southern Jiaobei Shoal (see Figure 3), all the isobaths moved seaward substantially with a silting process. In the western Jiaobei Shoal, the −2 m and −3 m isobaths retreated landward with a scouring process, but the −4 m and −5 m isobaths moved substantially seaward with a silting process. In the eastern shoal, the −2 m, −3 m, and −4 m isobaths retreated landward slightly with a scouring process, and the −5 m isobaths were stable. In the east of the mouth bar, all the isobaths moved seaward with a silting process. In the deep trough north of the mouth bar, the −2 m and −3 m isobaths turned westward slightly, and the −4 m isobaths extended seaward.

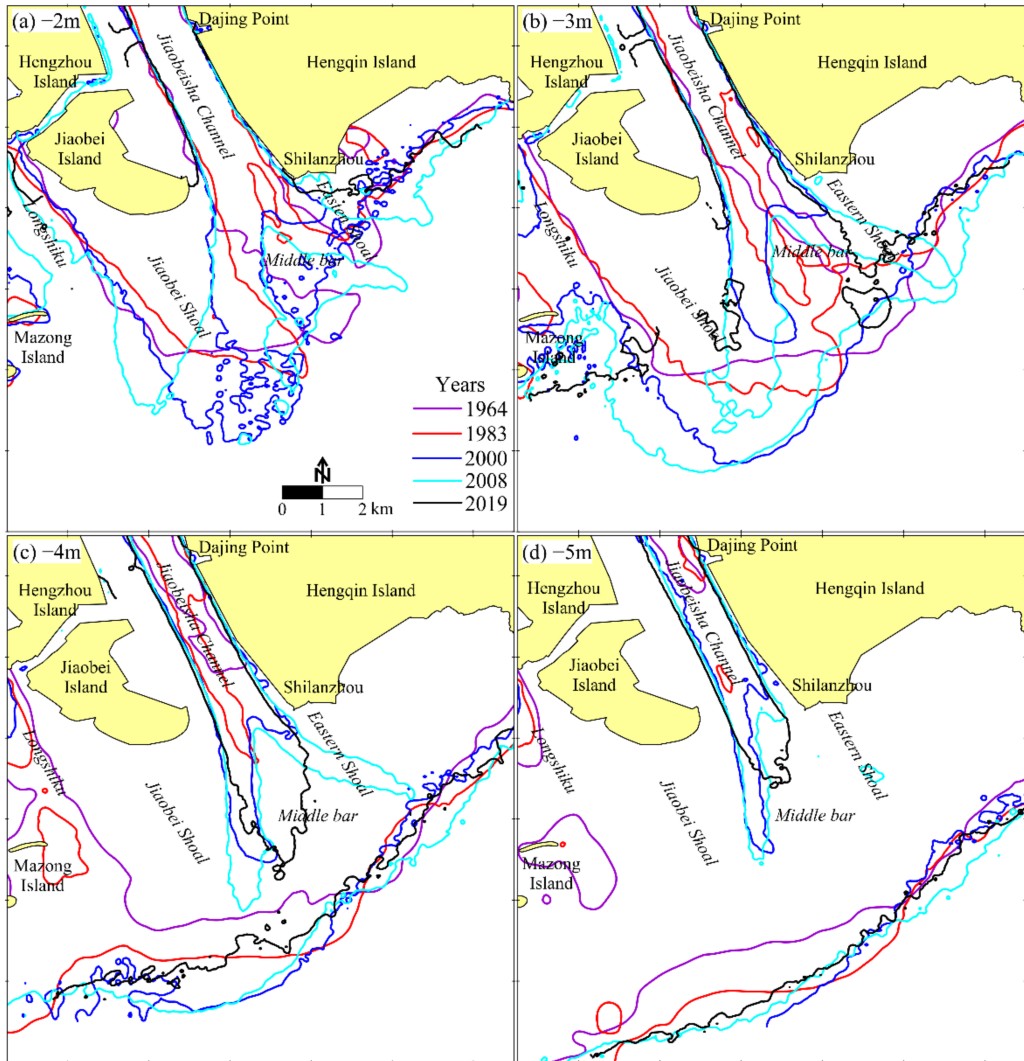

**Figure 3.** Planar changes of (**a**) −2 m, (**b**) −3 m, (**c**) −4 m, and (**d**) −5 m isobaths from 1964 to 2019.

From 1964 to 1983, the eastern and western Jiaobei Shoal and the deep trough of the Jiaobei Channel experienced a significant scouring process with a scouring range of 0.5–3.0 m (see Figure 4a); Longshiku, the east shoal, and southern Jiaobei Shoal experienced a significant silting process with a silting range of 0.5–2.0. There was slight siltation on the east side of the mouth bar.

Overall, the mouth bar maintained the geomorphic characteristics of "two shoals with one distributary branch" during this period; the mouth bar moved southward, and the distributary branch inside the mouth bar began to deflect westward.

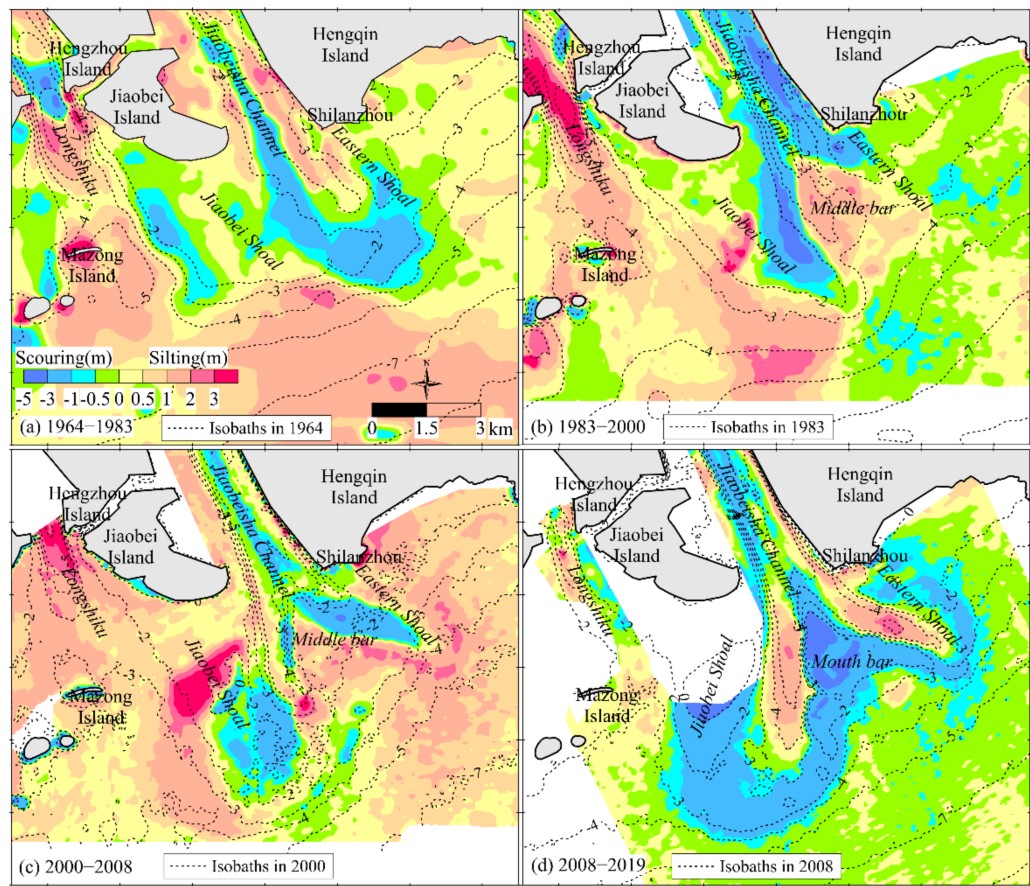

**Figure 4.** Planar maps of scouring and siltation in different decades. (**a**) from 1964 to 1983; (**b**) from 1983 to 2000; (**c**) from 2000 to 2008; (**d**) from 2009 to 2019.

4.1.2. Planar Changes from 1983 to 2000

According to the underwater topography in 1994 and 2000 (Figure 2c,d), the landform of "three shoals with two branches" appeared outside the river mouth of the Modaomen Estuary. The west branch was the main branch with an extending direction of SSE, and the east branch was the minor branch with an extending direction of ESE. A middle shoal connected with the east shoal formed between the two branches, and the water depth at the top of the middle shoal was less than −2 m. In 2000, the Jiaobeisha Channel and the west branch deepened with a water depth of below −5 m, and the west branch deflected to the southeast. The water depth of the shoal at the south of the west branch was less than −2 m. A crescent-shaped sandbar with an elevation above 0 m first appeared in the southern Jiaobei Shoal.

From 1983 to 2000, the −2 m and −3 m isobaths in the eastern and western Jiaobei Shoal moved westward significantly, and all the isobaths moved southward with a silting process in the southern shoal (see Figure 3). The −2 m, −3 m, and −4 m isobaths in the trough near Shilanzhou moved seaward and formed the east branch. The −2 m and −3 m isobaths in the west branch moved westward and those of −4 m and −5 m extended southward. All the isobaths to the east of the mouth bar were stable. The −2 m and −5 m isobaths of the Jiaobeisha Channel widened significantly, and the −5 m isobaths connected with those upstream, indicating that the deep trough of the Jiaobeisha Channel had widened and deepened.

From 1983 to 2000, the Jiaobeisha Channel, the western Jiaobei Shoal, and the south of Shilanzhou experienced a significant scouring process with a scouring range of 1.0–3.0 m and more than 5 m in parts (see Figure 4b). The central and southern Jiaobei Shoal, the middle shoal and Longshiku experienced a significant silting process with a silting

range of 0.5–3.0 m. There was slight scouring on the eastern edge of the mouth bar with a scouring range of 0.5–1.0 m.

Overall, the landform of the mouth bar changed obviously to the geomorphic characteristics of "three shoals with two distributary branches" during this period. The mouth bar and the west branch moved southward, and the east branch appeared southward of Shilanzhou with an extending direction of ESE. A middle shoal between the two branches was obviously silting up and that began to connect to the east shoal.

### 4.1.3. Planar Changes from 2000 to 2008

According to the underwater topography in the years of 2005 and 2008 (Figure 2e,f), the mouth bar maintained the geomorphic characteristics of "three shoals with two distributary branches". The west and east branches extended seaward with a SSE direction, the east branch deepened significantly, and the middle distributary shoal was separated from the east shoal. The area of the sand bar above 0-m isobaths in the southern Jiaobei Shoal increased significantly compared to that in 2000, and the sand bar gradually moved northward.

From 2000 to 2008, the −2 m isobaths in the Jiaobei Shoal, the middle shoal and the east shoal separated completely (see Figure 3). The −2 m, −3 m, and −4 m isobaths in the outer shoal of the east branch and the western Jiaobei shoal moved seaward. The −3 m, −4 m, and −5 m isobaths on the outer edge of the middle shoal retreated landward slightly. The −3 m and −4 m isobaths in the west and east branches extended considerably to the sea, and the −5 m isobaths in the west branch changed insignificantly. The −5 m isobaths on the southern edge of the east shoal moved seaward and changed a little in the Jiaobei Shoal.

From 2000 to 2008, the Jiaobeisha Channel, the southern Jiaobei Shoal, and the east branch experienced a significant scouring process with a scouring range of 0.5–3.0 m (see Figure 4c), and local scouring occurred on the southern edge of the middle shoal and the west branch. The northern and western Jiaobei Shoal, middle shoal, east shoal, and Longshiku experienced a significant silting process with a silting range of 0.5–3.0 m.

Overall, the landform of the mouth bar maintained the geomorphic characteristics of "three shoals with two distributary branches" during this period. The west and east branch extended seaward and the trough deepened. The mouth bar silted and expanded outward along the branches, and the middle shoal between the two branches silted and became shallower.

### 4.1.4. Planar Changes from 2008 to 2019

According to the underwater topography in 2010, 2016, and 2019 (Figure 2g–i), the mouth bar still maintained the geomorphic characteristics of "three shoals with two distributary branches" in 2010, but had significant changes in 2016 and 2019. The two branches and the middle shoal disappeared, with the water depth advancing from −3.5 to −4 m in the middle bar. The water depth in the southern Jiaobei Shoal and the east shoal all increased significantly and deep pits with a water depth greater than −5 m also appeared in the southern Jiaobei Shoal.

From 2008 to 2019, the −2 m and −3 m isobaths in the Jiaobei Shoal and the east shoal retreated significantly (see Figure 3), and the −2 m isobaths in the middle shoal completely disappeared, with the −3 m isobaths remaining. The −4 m isobaths in the deep trough in the north of the mouth bar were no longer branched with an increasing width, and the −5 m isobaths retreated. The −4 m isobaths outside the mouth bar retreated greatly, and the −5 m isobaths retreated slightly.

From 2008 to 2019, the shallow area of the mouth bar above −3 m experienced a significant scouring process with a water depth range of 1–5 m (see Figure 4d). The original west and east branches experienced a silting process with a silting range of 1–3 m. There was little change in the area of the mouth bar below −3 m.

Moreover, the geomorphic form of the mouth bar changed substantially during this period, and in a large area the shallow shoals above −3 m disappeared, and the two branches and the middle shoal no longer existed. The geomorphic form returned to "two shoals with one distributary branch" with the water depth increasing significantly. The topography around the mouth bar changed slightly during the period, indicating that the scouring sediment from the mouth bar did not accumulate in the surrounding area. Therefore, it is worth noting that the topographic change of the mouth bar during this period was very different from before.

### 4.2. Water Depth and Volume Changes of the Mouth Bar

According to the above analysis, there were considerable differences in morphological evolution characteristics of the mouth bar in the Modaomen Estuary in different decades. Therefore, according to the long-term bathymetric data from 1964 to 2019, calculations were made of the water depth and shoal volume (above −6-m isobaths) in different computed areas of the mouth bar (see Figure 1 for the location), and the mean water depth and the water volume (below 0 m isobaths) of the southern Jiaobeisha Channel. The research results are shown in Figure 5.

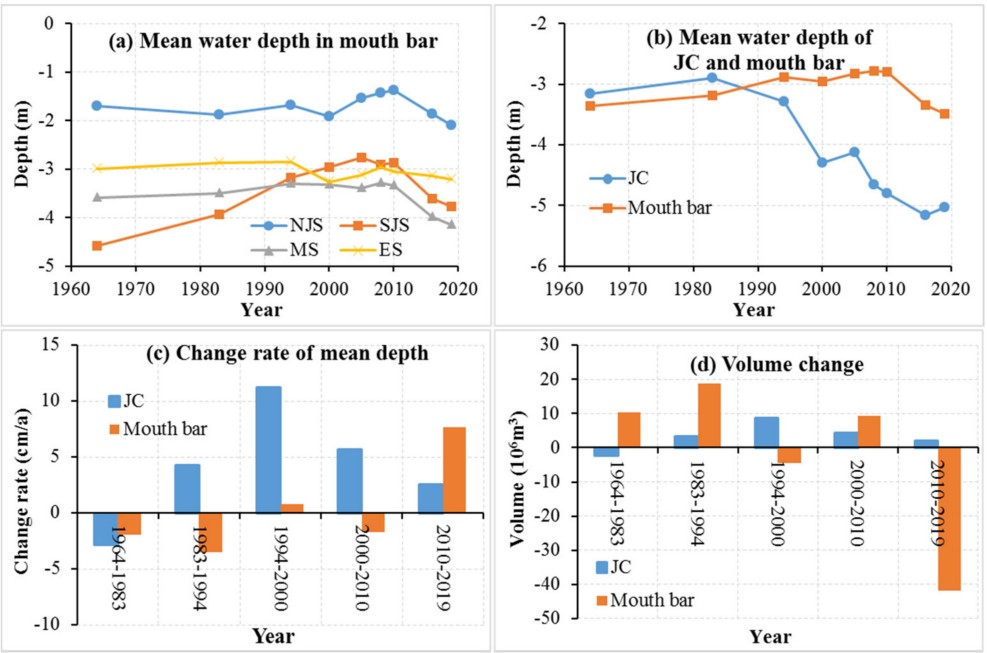

**Figure 5.** Mean depth and volume of the computed areas from 1964 to 2019. (**a**) mean water depth in the mouth bar, (**b**) mean water depth of JC and mouth bar, (**c**) change rate of mean depth; (**d**) Volume change of JC and mouth bar.

### 4.2.1. Water Depth Changes

In 1964, the mean water depth of the northern and southern Jiaobei Shoal, middle shoal, and east shoal was −1.7, −4.6, −3.6, and −3.0 m, respectively and that of the whole mouth bar was about −3.4 m (Figure 5a). The mean water depth of the southern Jiaobeisha Channel was approximately −3.2 m, hence the Jiaobeisha Channel was slightly shallower than the mouth bar.

From 1964 to 2019, the change characteristics were contrasted in different decades (Figure 5a). In the northern Jiaobei Shoal, the water depth increased by 0.2 m from 1964 to 2000, decreased by 0.5 m from 2000 to 2010, and increased by 0.6 m from 2010 to 2019. In the southern Jiaobei Shoal, the water depth decreased continuously from 1964 to 2005 with a total change range of 1.8 m, it increased by 0.1 m from 2005 to 2010, and increased by 0.9 m from 2010 to 2019. In the middle shoal, the water depth decreased by 0.3 m from 1964 to

1994, had no change from 1994 to 2010, and increased by 0.8 m from 2010 to 2019. In the east shoal, the water depth decreased by 0.1 m from 1964 to 1994, increased by 0.4 m from 1994 to 2000, decreased by 0.2 m from 2000 to 2010, and increased by 0.2 m from 2010 to 2019.

Regarding the mouth bar (Figure 5b), the mean water depth decreased by 0.5 m from 1964 to 2000, decreased by 0.15 m from 2000 to 2010, and increased by 0.7 m from 2010 to 2019. In 2019, the mean water depth of the mouth bar was about −3.5 m, exceeding that in 1964. In the southern Jiaobeisha Channel (Figure 5b), the mean water depth decreased by 0.3 m from 1964 to 1983 and increased significantly from 1983 to 2019, with a change range of 2.3 m. Thus, the Jiaobeisha Channel deepened from 1964 to 2019 with the water depth increasing from −3.0 to −5.0 m.

According to the mouth bar data (Figure 5c), the water depth decreased from 1964 to 1983, 1983 to 1994, and 2000 to 2010 with a change rate of −1.9, −3.5, and −1.7 cm/annum, respectively. The water depth increased from 1994 to 2000 and 2010 to 2019 with a change rate of 0.8 and 7.7 cm/annum, respectively. Among them, the change rate from 2010 to 2019 was considerablly higher than that in other periods. In the Jiaobeisha Channel (Figure 5c), the water depth decreased from 1964 to 1983 with a change rate of −2.8 cm/annum and increased from 1983 to 2019 with a change rate of 2.6–11.2 cm/annum. In particular, the change rate from 1994 to 2000 was significantly higher than that in other periods.

### 4.2.2. Water Volume Changes

According to the mouth bar data (Figure 5d), the shoal volume above −6 m isobaths increased from 1964 to 1983, 1983 to 1994 and 2000 to 2010, with a change range of $10.4 \times 10^6$ m$^3$, $18.9 \times 10^6$ m$^3$ and $9.3 \times 10^6$ m$^3$, respectively. The shoal volume decreased from 1994 to 2010 and 2010 to 2019 with a change range of $-4.4 \times 10^6$ m$^3$ and $-41.7 \times 10^6$ m$^3$, respectively. The shoal volume of the mouth bar increased by $34.2 \times 10^6$ m$^3$ from 1964 to 2010; thus, the increasing volume before 2010 was less than the decreasing volume from 2010 to 2019.

For the Jiaobeisha Channel (Figure 5d), the water volume below 0 m isobaths decreased from 1964 to 1983, with a change range of $2.1 \times 10^6$ m$^3$; the water volume increased from 1983 to 2019 with a change range of $2.0$–$8.8 \times 10^6$ m$^3$. The change rate from 1994 to 2000 was significantly higher than that in other periods. The water volume of the channel increased by $16.4 \times 10^6$ m$^3$ from 1964 to 2010, and the increasing volume after 2010 was much less than that before 2010.

### 4.3. Long-Term Morphological Evolution Characteristics

According to the above analysis, the morphological evolution characteristics of the mouth bar in the Modaomen Estuary had great contrasts in different decades. From 1964 to 2010, the geomorphic form of the mouth bar had evolved from "two shoals with one distributary branch" to "three shoals with two distributary branches", and the mouth bar experienced a silting process with shoals expanding seaward. After 2010, the geomorphic form of the mouth bar had returned to "two shoals with one distributary branch" with wider troughs and deeper shoals, and the mouth bar experienced a scouring process with shoals deepening.

The extending direction of the distributary branch inside the mouth bar, which gradually shifted from SE to S from 1964 to 2000, changed only slightly from 2000 to 2010, and then shifted to SSE from 2010 to 2019. The distributary branch inside the mouth bar mainly extended S at the south of Shilanzhou, which revealed that the water and sediments were mainly transported southwards from the river mouth.

## 5. Discussion

### 5.1. Impact of Water and Sediment Inputs

As a fluvial-dominated estuary, sediment discharge from upstream is typically the most important input condition for mouth bar evolution. In recent years, upstream sediment loads have decreased due to climate change and human activities, such as dam construction and artificial afforestation among others. The silting rate of the estuarine topography has decreased, and some regions have faced scouring problems [4,5,8,20,25,30].

The sediment load input in the Modaomen Estuary accounted for approximately 35% of the total amount of the Xijiang River at Makou station [31]. Therefore, sediment discharge from the Xijiang River, the main tributary of the Pearl River, plays a key role in the evolution of the mouth bar of the Modaomen Estuary. The annual runoff and sediment load of the Xijiang River (Makou station) changed slightly from the 1960s to the 1990s, with mean annual amounts of $230 \times 10^9$ m$^3$ and $73 \times 10^6$ t, respectively. The annual sediment load of the Xijiang River decreased by 70% after the 1990s to the present, but the runoff changed slightly. The mean annual amount of sediment load was about $22.7 \times 10^6$ t in the last 20 years (Figure 6), which was mainly caused by dam construction in the river basin rather than a precipitation change [32]. Therefore, the sediment load in the Modaomen area also decreased sharply after 2000. However, the annual silting rate of the mouth bar of the Modaomen Estuary from 2000 to 2010 was the same as that from 1964 to 1989. Although the sediment load from upstream was the same before and after 2010, the evolutionary characteristics were very different, and there was a significant scouring in the mouth bar after 2010. Thus, the evolutionary characteristics of the mouth bar in recent decades could not have been caused by the sediment load change from the Xijiang River.

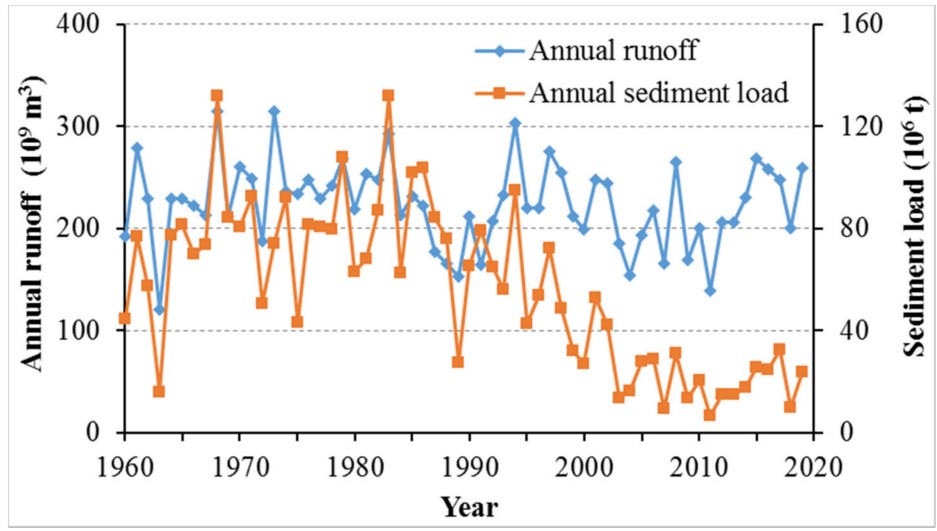

**Figure 6.** Changes in the annual runoff and sediment load at Makou station since 1960.

In late June of 2005, the Xijiang River experienced a 100-year flood. The topography of the mouth bar in September of 2005 showed that the west and east branches were greatly extrapolated, the east branch almost rushed out of the bar, and the mouth bar expanded seaward. Therefore, the mouth bar which extended significantly into the sea, was related to the peak flood of the Xijiang River.

### 5.2. Impact of Human Activities

Many different human activities appearing frequently in estuarine regions include estuarine regulation, shoal reclamation, navigating channel regulation, and sand dredging, and these can change the estuarine boundary and topography, and affect the dynamic and geomorphic form of the mouth bar [22,26,30,31]. In recent decades, the main human activities in the Modaomen Estuary have been estuarine reclamation and sand dredging [22,28].

5.2.1. Impact of Shoal Reclamation in the West Shoal

Before the 1980s, the river mouth of the Modaomen Estuary connecting the Modaomen Channel upstream was located near Guading Point (Figure 7), and south of the Guading Point was a shallow sea, which was covered by rock islands such as Hengqin Island, Sanzao Island, Shilanzhou, Mangzhou, Hengzhou, and Hezhou among others. The geomorphic form of the shallow sea was also "three shoals with two troughs". The three shoals, from west to east, were the Sanzao Bay shoal, the Hezhou–Hengzhou shoal and the Hengqin shoal, and the two troughs were Longshiku and the Jiaobeisha Channel. A small-scale river mouth had developed outside Shilanzhou and Hengzhou.

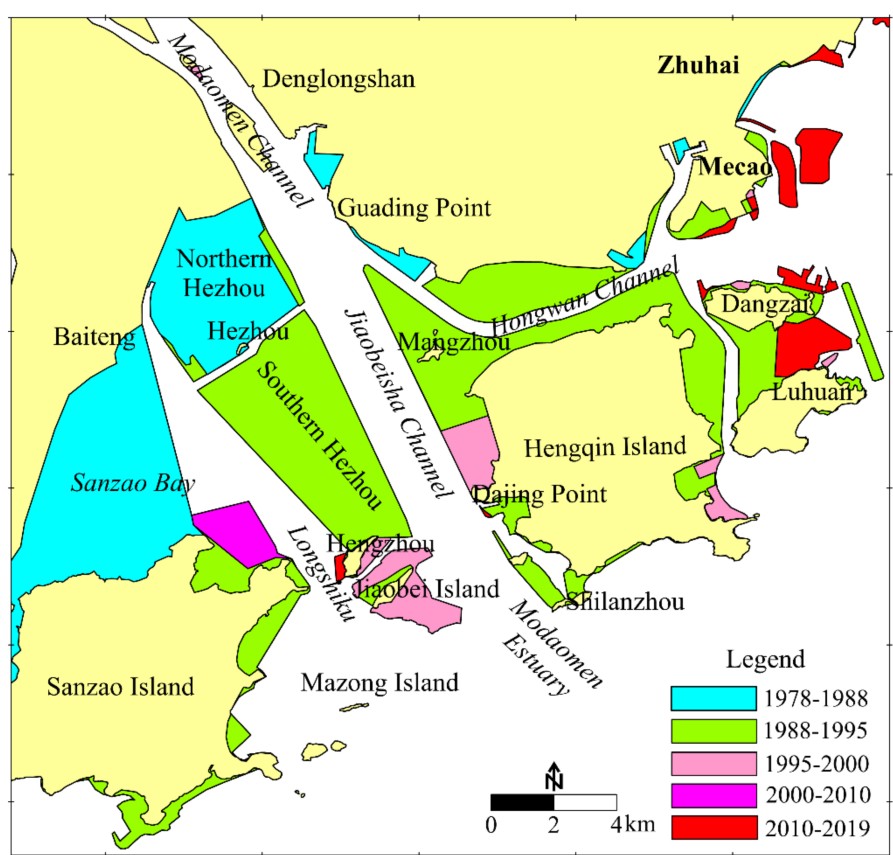

**Figure 7.** Planar map of the reclamation area during different decades.

From the 1980s to the mid-1990s, with the implementation of shoal reclamation and estuarine regulation in Sanzao Bay, Hezhou–Hengzhou shoal, and Hengqin Shoal (Figure 7), the shallow sea of the Modaomen Estuary gradually evolved into two channels into the sea, which were controlled by a fixed boundary. The Jiaobeisha Channel was the main channel accounting for 85% of the runoff and sediment load from the Modaomen Channel; the Hongwan Channel was the branch flowing into the shallow sea area of Macao. Longshiku was converted into a tide-dominated estuary with little connection with the Jiaobeisha Channel.

When the river mouth of the Modaomen Estuary moved from Guading Point to Shilanzhou, the Jiaobeisha Channel continuously eroded and deepened under strong runoff, and the scouring sediment was transported to the mouth bar. The mouth bar near Shilanzhou had a strong interaction combining with strong runoff and ocean dynamics, and became the accumulating center of the estuary. Therefore, the reclamation activities in the Modaomen Estuary from the 1960s to the early 21st century played an important role in the significant siltation process of the mouth bar. After 2000, the reclamation activities in the Modaomen Estuary stopped, while the siltation change in the barrage area was not

obvious from 2000 to 2010. After 2010, the significant increase in the water depth in the mouth bar should have had little relation to the topographic changes of the Modaomen shoreline boundary caused by reclamation.

### 5.2.2. Impact of Sand Dredging Activity

In recent years, sand dredging activities in the estuarine region became an important human activity, especially in the well-developed PRD region [9,33,34]. Sand dredging activities in the mouth bar of the Modaomen Estuary had been reported from the 1990s to 2010 [22,28] as one of the causes why the silting rate was not obvious from 2000 to 2010. According to the local river management authority, although sand dredging activities in the mouth bar had converged after 2010, there were high-intensity sand dredging activities in the mouth bar, mainly serving the shoal reclamation activities in the surrounding areas.

Under natural conditions, the top of the mouth bar was significantly affected by waves, and the bed sediment was mainly well-sorted fine sand, which was a high-quality sand source for reclamation and construction. The sediment samples of 2006 showed that the well-sorted fine sand and silty sand mainly distributed at the top of the mouth bar (Figure 8a), with a medium grain size (d50) of 0.06–0.15 mm. Poorly sorted clayey silt mainly distributed in the Jiaobeisha Channel and the outer slopes of the mouth bar, with a d50 mostly less than 0.03 mm. After high-intensity sand dredging activities, the water depth in the mouth bar increased significantly, and the influence of wave action was reduced significantly, so the bed sediment in the mouth bar was refined. The sediment samples of 2019 showed that the poorly-sorted clayey silt was mainly distributed on the mouth bar with a d50 mostly less than 0.03 mm (Figure 8b).

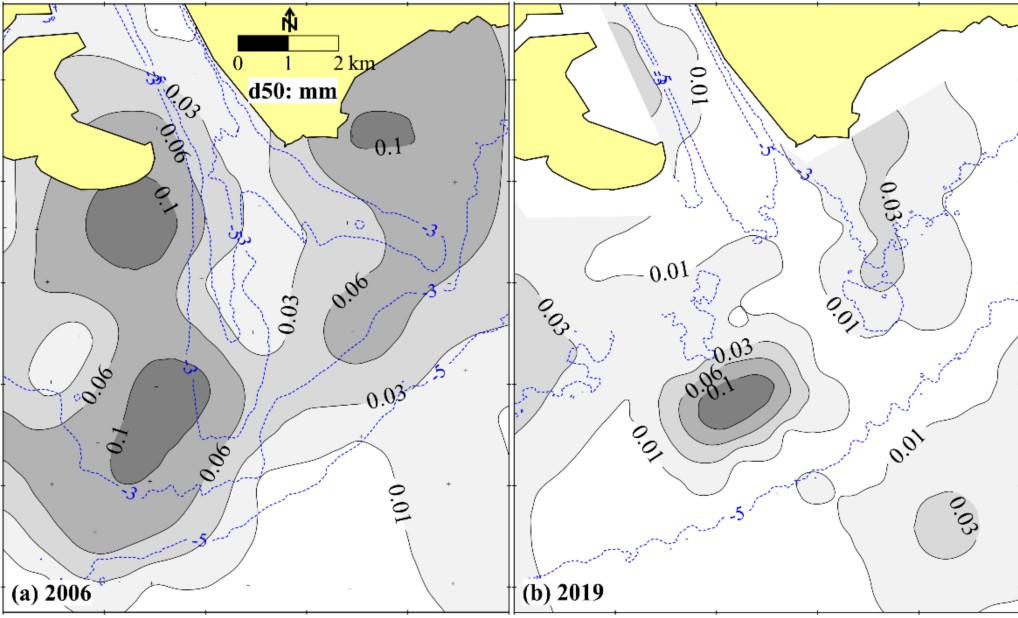

**Figure 8.** Medium grain size (d50) at the mouth bar measured in 2006 (**a**) and 2019 (**b**).

### 5.3. Impact of Ocean Hydrodynamics

The ocean dynamics affecting the mouth bar mainly include tides and waves [3]. Tidal range in estuarine areas is an important index to measure the strength of tidal dynamics. The mean annual tidal range of Sanzao station outside the mouth bar of the Modaomen Estuary changed slightly from the 1970s to 2010s. The mean annual tidal range of Dahengqin station inside the mouth bar decreased by approximately 0.15 m from 1980 to 2000, and increased slightly after 2000. The decreasing trend of the tidal range at Dahengqin station from 1980 to 2000 indicated that the fluvial power of the Jiaobeisha Channel increased significantly after the Modaomen Estuary regulation.

Wave dynamics could play an important role in sediment movement and geomorphic shaping in the mouth bar area. The dominant wave direction in the Modaomen Estuary was SE. With the continuous silting process, the mouth bar became shallower and faced stronger wave dynamics on the sediment transport of the mouth bar, resulting in sedimentary characteristics dominated by well-sorted sandy sediments at the top of the mouth bar. Moreover, wave action could help sediment accumulation on top of the mouth bar and make the mouth bar shallower, also pushing the water flow southwards.

In addition, under the action of SE waves, the eroded sediments in the east of the mouth bar may gradually transport westward and deposit in the southern Jiaobei Shoal, which was one of the reasons for the silting process in the southern Jiaobei Shoal before 2010.

*5.4. Implications for Sustainable Management*

The Modaomen Estuary is the main flow path of the Pearl River, especially in flood season. Its management and development are not only related to flood control and disaster reduction, but also related to the protection of the estuarine environment and the water supply safety for Zhuhai, Zhongshan, and Macao, which are important cities in the Guangdong–Hong Kong–Macao Great Bay Area. In the past decades, with the implementation of shoal reclamation and estuarine regulation in the Modaomen Estuary, the geomorphic form has undergone significant changes and the river mouth has substantially transferred seaward. The Jiaobeisha Channel has deepened gradually, and the mouth bar has changed significantly. The evolution and management of the mouth bar in the Modaomen Estuary has involved many scientific issues, such as tidal dynamic change, saltwater intrusion, and navigating channel regulation among others, which have concerned many scholars [24–28].

With economic development, the existing waterway of the Xijiang River, going to the open sea through the Modaomen–Hongwan Channel or Hutiaomen Channel–Huangmao Bay, cannot meet the growing shipping demand. Thus, the navigating regulation of the mouth bar in the Modaomen Estruary has become a key scientific problem, requiring urgent research. Due to the abundant sediment coming from the Pearl River, the direct excavating trench for the navigating channel in the mouth bar may face serious back siltation. If a regulating dike was constructed for reducing the sediment siltation of the excavating trench at the mouth bar, it would face flood disaster and estuarine changes of topography. Excavating a trench may cause saltwater intrusion, which would affect local water supply safety and the water environment.

Recently, the morphological evolution has become a key problem after the large-scale deepening of the mouth bar by sand dredging activities. Against the background of a significant reduction of sediment load from the Pearl River, further research using additional field bathymetric data and hydrological data is required to identify whether the mouth bar will recover to the geomorphic form before 2010 in a short time.

**6. Conclusions**

This study explored the long-tern morphological evolution of the mouth bar in the Modaomen Estuary of the Pearl River from 1964 to 2019. In the last 55 years, due to the impact of human activities, such as shoal reclamation and estuarine regulation in the Modaomen Estuary, the river mouth moved out of the shallow sea covered by several rock islands and faced the South China Sea directly. Therefore, the mouth bar became a siltation center in the estuarine region and expanded outwards, gradually evolving into a geomorphic pattern with three shallow shoals and two distributary branches; a west branch as the main branch accompanied by a small east branch. Over the past decade, high-intensity sand dredging activities in the mouth bar have led to a considerable deepening of the water depth and a significant refinement of bed sediments, forming a discharge pattern of a wide and shallow channel flowing into the sea. Therefore, the evolutionary characteristics of the mouth bar are different and have become abnormal in recent years.

In addition, the Modaomen Estuary is the main flow path of the Pearl River, especially in the flood season, so the evolution of the mouth bar relates to the disaster prevention of floods and storms, navigating channel regulation, water supply safety, environment protection, utilization of water resources, and the sustainable development of the PRE, among others. More field bathymetric data and hydrological data are needed to carry out further research work on the subsequent evolution of the mouth bar and its impact, especially against the background of a significant reduction of sediment discharge and high-intensity activities.

**Author Contributions:** Writing—original draft, Z.H.; methodology, H.L.; writing—review and editing, H.X.; funding acquisition, B.Y. and M.X. All authors reviewed the manuscript. All authors have read and agreed to the published version of the manuscript.

**Funding:** This research is funded by the National Key Research and Development Program of China (Grant No. 2019YFB1600605); the National Natural Science Foundation of China (Grant No. 52171260) and the Innovation Fund of Tianjin Research Institute for Water Transport Engineering (Grant No. TKS20210104).

**Institutional Review Board Statement:** Not applicable.

**Informed Consent Statement:** Not applicable.

**Data Availability Statement:** Not applicable.

**Acknowledgments:** The authors thank the editor and anonymous reviewers whose invaluable and constructive suggestions greatly improved the scientific quality of the original manuscript. The authors also thank Editage for its linguistic assistance during the preparation of this manuscript.

**Conflicts of Interest:** The authors declare no conflict of interest.

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
