# Peer review of "Long-Term Geomorphological Evolution of the Mouth Bar in the Modaomen Estuary of the Pearl River over the Last 55 Years (1964–2019)"

_water, doi:10.3390/w14010090_

Round 1

Reviewer 1 Report

The manuscript presents an interesting effort to analyze the morphological evolution of a mouth bar.  I like the depth of the analysis and the insights gained from it.  

I also appreciate the authors' attempt to capture the implications from the morphological changes happening to the river system. 

The primary concerns I have are listed below:

1) Overall, the English language of the manuscript must be carefully reviewed and improved. There are grammatical and sentence-structure issues throughout the manuscript.  It did not hinder my ability to review the technical value of the manuscript, but it does take away from the ability to smoothly read through it. 

2) The abstract, in particular, fails to highlight the exact contribution of the paper. The authors must rewrite the abstract to ensure that the contributions are clearly and directly stated.  

Author Response

Dear Reviewer:
Thank you for your work and the comments concerning our manuscript entitled “Long-term morphological evolution of the mouth bar in Modaomen Estuary of the Pearl River over the last 55 years(1964–2019)” (ID: Water-1505632). Those comments are all valuable and very helpful for revising and improving our paper. We have studied the comments carefully and have made correction which we hope meet with approval. The main corrections in the paper and the responds to the reviewer’s comments are as flowing. 
(1) Response to comment: Overall, the English language of the manuscript must be carefully reviewed and improved. There are grammatical and sentence-structure issues throughout the manuscript. It did not hinder my ability to review the technical value of the manuscript, but it does take away from the ability to smoothly read through it. 
Response: Thank you for your suggestion. Actually, it is really difficult for a Chinese researcher to write in English, so my origin manuscript had been improved by Editage (www.editage.cn) for the linguistic assistance. If necessary, I will polish the revised manuscript again by Editage. 
(2) Response to comment: The abstract, in particular, fails to highlight the exact contribution of the paper. The authors must rewrite the abstract to ensure that the contributions are clearly and directly stated.
Response: Thank you for your suggestion, the “abstract” has been rewritten. Please it again and I look forward to your valuable suggestions. 

Reviewer 2 Report

Dear Editor,

Thank you for giving me the opportunity to review this MS. There are no issues of conflicting interest, and I have no personal or professional affiliation with the authors.

This manuscript presents a study on Long-term morphological evolution of the mouth bar in Modaomen Estuary of the Pearl River over the last 55 years (1964–2019).

This is a potentially good manuscript. However, I appreciate to take these comments into account.

The following revisions are suggested:

General comments:

  • Title: may you can change "morphological" to "geomorphological"
  • It is better to support the study with more data, as annual precipitation, land-use maps (in addition to reclamation that you used), if it is possible.

Abstract:

  • Line 15: “The following results were obtained” delete this sentence.

Figures:

  • (3): on the figure you wrote (a)2m, (b)3m,….. add (-) on the figure as in caption.

Author Response

Dear Reviewer:
Thank you for your work and the comments concerning our manuscript entitled “Long-term morphological evolution of the mouth bar in Modaomen Estuary of the Pearl River over the last 55 years(1964–2019)” (ID: Water-1505632). Those comments are all valuable and very helpful for revising and improving our paper. We have studied the comments carefully and have made correction which we hope meet with approval. The main corrections in the paper and the responds to the reviewer’s comments are as flowing. 
(1) Response to comment: Title: may you can change "morphological" to "geomorphological".
Response: As your suggestion, "morphological" has been revised to “geomorphological” in the title. 
(2) Response to comment: It is better to support the study with more data, as annual precipitation, land-use maps (in addition to reclamation that you used), if it is possible.
Response: Usually, annual precipitation and land-use changes in the river basin may play an important role in the changes of runoff and sediment discharge of the river. But in the Pearl River of China, the sediment discharge decreased considerably in recent years was mainly caused by dam construction, which has been added in “5.1 Impact of water and sediment inputs” 
(3) Response to comment: Line 15: “The following results were obtained” delete this sentence.
Response: As your suggestion, the sentence has been deleted.
(4) Response to comment: on the figure you wrote (a)2m, (b)3m,….. add (-) on the figure as in caption.
Response: As your suggestion, the figure 3 has been revised.

Round 2

Reviewer 1 Report

I am satisfied with the revision of the manuscript. 

Reviewer 2 Report

Dear Editor,

The authors have addressed all my comments in this manuscript.

Therefore, I recommend publishing it in present form.